# TinyLUT: Tiny Look-Up Table for Efficient Image Restoration at the Edge

**Huanan Li**[1,2], **Juntao Guan**[1,2,3], **Rui Lai**[1,2*], **Sijun Ma**[1,2], **Lin Gu**[4,5*], **Zhangming Zhu**[1,2]

[1]Key Laboratory of Analog Integrated Circuits and Systems (Ministry of Education)
[2]School of Integrated Circuits, Xidian University, Xi'an, China
[3]Hangzhou Institute of Technology, Xidian University, Hangzhou, China
[4]RIKEN AIP, Tokyo103-0027, Japan
[5]The University of Tokyo, Japan
{huananli,sijunma}@stu.xidian.edu.cn
{guanjuntao,zhangmingzhu}@xidian.edu.cn
rlai@mail.xidian.edu.cn, lin.gu@riken.jp

## Abstract

Look-up tables(LUTs)-based methods have recently shown enormous potential in image restoration tasks, which are capable of significantly accelerating the inference. However, the size of LUT exhibits exponential growth with the convolution kernel size, creating a storage bottleneck for its broader application on edge devices. Here, we address the storage explosion challenge to promote the capacity of mapping the complex CNN models by LUT. We introduce an innovative separable mapping strategy to achieve over $7\times$ storage reduction, transforming the storage from exponential dependence on kernel size to a linear relationship. Moreover, we design a dynamic discretization mechanism to decompose the activation and compress the quantization scale that further shrinks the LUT storage by $4.48\times$. As a result, the storage requirement of our proposed TinyLUT is around 4.1% of MuLUT-SDY-X2 and amenable to on-chip cache, yielding competitive accuracy with over $5\times$ lower inference latency on Raspberry 4B than FSRCNN. Our proposed TinyLUT enables superior inference speed on edge devices with new state-of-the-art accuracy on both of image super-resolution and denoising, showcasing the potential of applying this method to various image restoration tasks at the edge. The codes are available at: https://github.com/Jonas-KD/TinyLUT.

## 1 Introduction

With image sensors widely used in IoT applications, the demand for image restoration on edge devices has been stimulated. For decades, classic methods including interpolation [1, 2], sparse coding [3, 4, 5] and domain transformation [6] have been proposed. In recent years, deep learning based algorithms [7, 8, 9, 10, 11] have made encouraging progress in restoration quality, while suffering from high computational load and long inference latency. These issues limit their extensive applications in resource-constrained IoT terminals. Thus it can be seen, the growing demand for edge devices (*e.g.* smartphones and Raspberry Pi) calls for an extremely lightweight solution for image restoration with high quality. Given this, researchers develop LUT-based methods [12, 13, 14] to supplant the intensive convolutional computations with direct memory access operation for accelerating the inference, and exhibiting enormous potential.

In recent research, SRLUT [15] initially transfers the output values of a small trained deep network with receptive field (RF) size 4 to uniformly sampled LUT. To further improve the super resolution

---

*Corresponding author

38th Conference on Neural Information Processing Systems (NeurIPS 2024).

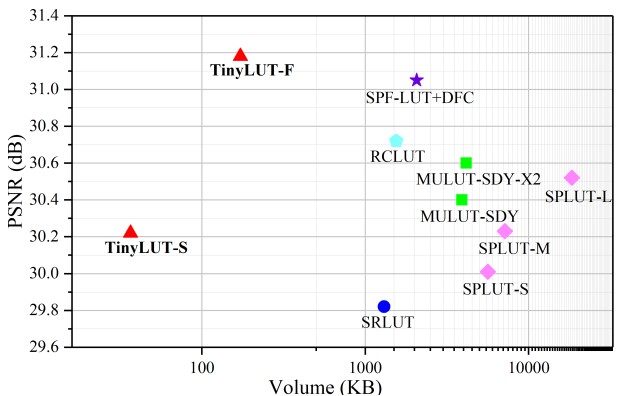

Figure 1: Performance comparison of our TinyLUT and other state-of-the art LUT-based algorithms. PSNR $vs.$ Volume on Set5 benchmark dataset for $\times 4$ super resolution task. TinyLUT outperforms the competitors in accuracy with over $10\times$ lower memory consumption.

Table 1: LUT size estimation when storing 8bit data with output entities $r = 4$. Compared with the full size, the sampled operation effectively reduces the storage consumption of LUT. Furthermore, separable mapping strategy(SMS) and trained dynamic discretization mechanism (DDM) achieve an exaggerated reduction result.

| RF | LUT | Full size | SRLUT | SMS | SMS + DDM |
|---|---|---|---|---|---|
| $1 \times 1$ pixels | 1D | 4KB | 272B | 4KB | 0.89KB |
| $2 \times 2$ pixels | 4D | 64GB | 1.27MB | 16KB | 3.57KB |
| $3 \times 3$ pixels | 9D | 64ZB | 1767GB | 36KB | 8.04KB |

quality, Li et al. [14] increases the RF size through the different kernel shape combination with multiple LUTs. Meanwhile, Ma et al. [13] constructs a series-parallel LUT framework based on multiple LUTs cascade to improve the prediction accuracy. However, the existing LUT-based schemes face the challenge of storage explosion when mapping 8bit image with larger than $2 \times 2$ kernel size. The storage explosion invalidates previous methods where the required storage of LUT grows exponentially as the number of indexing entries (*i.e.* input entities) increase. As reported in Table 1, $3 \times 3$ kernel, commonly used in CNNs, will need a tremendous storage up to $(2^8)^9 \times 8\text{bit} \times 16 = 64\text{ZB}$ and $(2^4 + 1)^9 \times 8\text{bit} \times 16 = 1767\text{GB}$ in full LUT and uniformly sampled LUT [15] with 8bit data respectively. Therefore, the issue of storage explosion presents severe challenge when mapping the complex models by LUT on edge devices.

To overcome the storage explosion challenge, we propose the separable mapping strategy(SMS) and dynamic discretization mechanism (DDM) to decouple the kernel and activation, respectively. As reported in Table 1, the reduction of LUT storage reaches exponential levels using SMS and further achieves a $4.48\times$ times reduction by DDM when the RF size is $3 \times 3$ and storing 8bit data.

From the perspective of kernel decomposition, considering the exponential relationship between the dimension of input entities and LUT size, we propose the innovative SMS to decouple the convolution kernel by spatial location relationship. SMS decomposes the convolution weight of $n$ input entities into the parallelization of $n$ component independent sub-operations, which are instantiated as $1 \times 1$ kernel size respectively to reduce the number of input entities in LUT-based inference, consequently reducing the dimension of $n$D LUT. Then according to the additivity of convolutions with compatible kernel sizes[16], we reconstruct the feature space by averaging each output of $n$ parallel layers. Finally, each branch with $1 \times 1$ kernel are individually mapped by 1-dimension LUT (1D LUT) with 1 input entity for $s$ possible values. This approach reduces the size of $n$D LUT from $s^n$ to $s \times n$ and allows vanilla convolution to be LUT-mapped on resource-constrained devices.

As for the activation compression, previous methods [15, 14, 17] uniformly sample the input values to reduce the possible values of $s$ and apply interpolation to recover the inference accuracy. We analyze the advantages of SPLUT [13] and propose dynamic discretization mechanism (DDM) to decompose the activation into Most Significant Bits (MSBs) and Least Significant Bits (LSBs). Moreover, inspired by the activation quantization methods[18, 19], DDM introduces the learnable clipping parameters to explicitly compress the quantization scale of MSBs and LSBs activation, thereby reducing the possible pixels values and decreasing LUT scales.

As illustrated in Fig 1, we take the classic image restoration task single image super resolution (SISR) as an example. Our proposed TinyLUT-F consumes only 4.1% storage of MuLUT-SDY-X2[14] to achieve over $0.58$dB PSNR increase in Set5 testset. Compared to other LUT-based methods, TinyLUT significantly reduces storage overhead while improving accuracy, expanding the application of LUT-based approaches on edge devices.

The main contributions can be summarized as follows:

- We reveal the scheme of storage explosion, being the key problem that limits the application of LUT in edge devices. Focus on this, we propose an innovative separable mapping strategy(SMS) to realize dimensional reduction of LUT and solve the storage explosion problem.

- We design a dynamic discretization mechanism(DDM) to decompose the activation and compress the corresponding quantization scale, further shrinking the storage requirement without considerable accuracy loss.

- Our comprehensive experiments demonstrate that TinyLUT, constructed by SMS and DDM, achieves significant restoration accuracy over previous LUT methods with minimal storage consumption, showing its potential for application on edge devices.

## 2   Related Works

**Deep Learning for Image Restoration:**   In recent years, image restoration performance has been rapidly developed through deep learning methods [20, 8, 21, 22, 10, 7, 11, 23, 24, 12, 25, 26] and achieved outstanding accuracy. The authors of [24] presented a super resolution method using very deep networks and residual learning. Zhang et al. [7] proposed a deep convolutional neural network with residual learning to handle image restoration tasks such as denoising and SISR. FFDNet [10] achieves more efficient and flexible than DnCNN [7] by down-sampled method. Also it have the ability to robustly control the trade-off between noise reduction and detail preservation. As a fast and memory-efficient network, FMEN [23] is constructed by enhanced residual block and high-frequency attention block. Dong et al. [8] explored a more efficient network by re-designing the SRCNN structure. In [27] author proposed a deep convolutional network within a Laplacian pyramid framework for fast and accurate image super-resolution. Denoising results of FastDVDnet [28] feature remarkable temporal coherence, very low flickering, and excellent detail preservation. However, these CNN methods suffer form intensive computations of convolutional operator, which limits their deployment on edge devices.

**Lookup Table for Image Restoration:**   LUT-based methods are commonly used in hardware systems[29] and some embedded devices[30] to accelerate CNNs by pre-storing the results of complex functions with high efficient retrieval operations. In image restoration tasks, LUTs are commonly used in video coding [31, 32], tone-mapping [33] and image enhancement [17]. Furthermore, deep learning methods based on LUTs have also emerged in SISR[15, 14, 13] and gradually garnered attention. SRLUT[15] establishes a spatial mapping from a local RF block to the corresponding high-resolution pixel. In MuLUT[14] and SPLUT[13], the authors proposed the collaborating structure by multiple LUTs to increase the model's RF and improve the inference accuracy. Furthermore, Liu et al.[34] found that the interaction of space and channel features in vanilla convolution allows previous methods to increase the RF at the cost of linearly increasing the LUT size. They proposed RCLUT based on the Reconstructed Convolution(RC) module to enlarge the RF, achieving significant performance with less storage. Realizing the dilemma between the performance improvement and storage of LUT-based models, Li et al.[35] introduce the Diagonal-First Compression(DFC) framework to achieve a better trade-off between storage size and performance for image restoration. Additionally, they also designed a new structure, SPF-LUT, to further improve the performance of LUT-based models. However, the RF and channels of LUT-based methods are limited to adapt the trade-off between storage and accuracy. In comparison, our work is focused on developing a lightweight LUT-based framework to improve the image restoration accuracy.

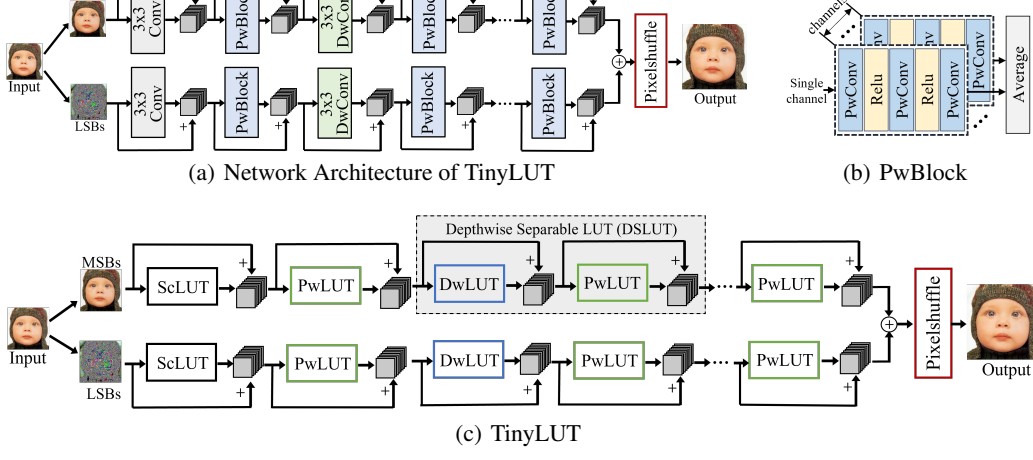

(a) Network Architecture of TinyLUT

(b) PwBlock

(c) TinyLUT

Figure 2: (a)The network overview of TinyLUT. (b)The structure of PwBlock. (c)The LUT framework overview of TinyLUT.

## 3 Method

### 3.1 Overview of TinyLUT

The structure of TinyLUT and foundational components are depicted in Fig 2. We design two parallel branches with cascaded LUTs to process Most Significant Bits (MSBs) and Least Significant Bits (LSBs) data respectively, and introduce separable mapping strategy and dynamic discretization mechanism.

According to the previous LUT-based methods we follow two simple design rules:(i) to improve model accuracy, the RF size needs to be increased; and (ii) the number and storage requirement of LUTs need to be reduced to adapt edge devices. Therefore, our CNN network (Fig 2(a)) is mainly inspired by the architecture of SRLUT [15] and DnCNN [7]. Specifically, the CNN model is built on standard convolution, depthwise convolution (DwConv) and PwBlock. The PwBlock 2(b) is a channel combining module based on SMS and described in Section 3.2. Based on SMS and DDM, the standard convolution, DwConv and PwBlock are mapped by LUT respectively to build standard convolution mapped LUT (ScLUT) , depthwise LUT (DwLUT) and pointwise LUT (PwLUT). In particular, we refer the philosophy of depthwise separable convolutions[36] to combine DwLUT and PwLUT to depthwise separable LUT(DSLUT). The mapping details for DwLUT and PwLUT are described in Section 3.2 and Section 3.3.

As in depthwise separable convolutions[36], the DwLUT optimizes the LUT size while maintaining a large effective RF. The PwLUT is used for cross-channel feature information integration and model channel number transformation. We introduce skip connections to fuse the real-value inputs and the retrieval outputs between DwLUT and PwLUT to avoid the degradation. The RF and output channels are set to $3 \times 3$ and 16. Due to each color channel having to be processed independently as in SRLUT [15], the input channel of ScLUT is set to 1.

According to the different number of DSLUT serial stacks, we build TinyLUT-F with 7 stacked. TinyLUT-S is built by ScLUT and double PwLUTs. As shown in Fig 2(c), TinyLUT is constructed by DwLUT and PwLUT in each branch. An additional PwLUT is added at the end of each branch to adjust the number of output channels of different image restoration task. In super resolution, the output channel number of the last PwLUT is $r^2$ to adapt to the pixelshuffle [21], where $r$ is the upscaling factor. In order to further expand the RF size, we introduce rotational ensemble trick [15] in the inference.

### 3.2 Separable Mapping Strategy

In Table 1, we provide an estimation of the LUT size of four output channels and single input. It is evident that the SRLUT continues to face the issue of storage explosion, resulting in approximately

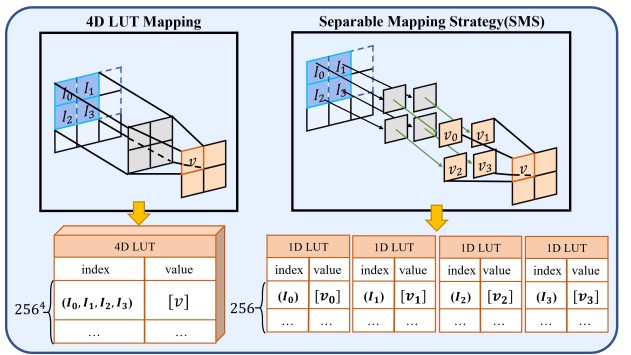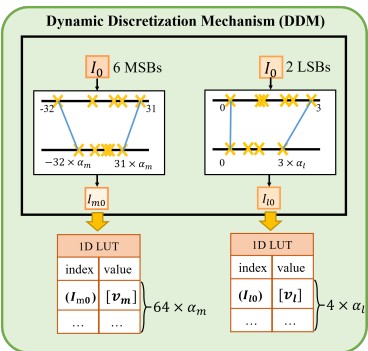

Figure 3: **Left**: Existing LUT-based method maps the $2 \times 2$ input convolution to a 4-dimensional LUT (4D LUT). SMS decomposes the convolution kernel to reduce the storage from $256^4 \times 8$bit to $256 \times 4 \times 8$bit. **Right**: The dynamic discretization mechanism (DDM) decouples the activation and compress the precision to reduce the storage to $(\alpha_m \times 64 + \alpha_l \times 4) \times 8$bit using $\alpha_m, \alpha_l \in [0, 1]$.

1767GB of memory overhead when processing $3 \times 3$ pixels of input simultaneously. In contrast, the proposed separable mapping strategy (SMS) alleviates the memory bottleneck associated with a large RF and reduces the required volume to just 36KB. Therefore, we enable the transfer of more deep convolutional network model into efficient LUT framework with fewer hardware cost.

The left image in Fig 3 illustrates 4-dimensional LUT(4D LUT) mapping the convolution $2 \times 2$ kernel with single output in existing methods [15, 14, 13]. It requires $256^4 \times 8$bit= 4GB and exceeds the storage capacity of edge devices. Therefore, we propose separable mapping strategy(SMS) to decouple $2 \times 2$ kernel into four $1 \times 1$ kernels which reduces the size of the LUT required to $256 \times 4 \times 8$bit= 1KB. Therefore, we propose the DwLUT. Firstly, from the original depthwise convolution operation, it can be obtained by:

$$F_{out} = \sum_{i=0}^{n-1} \sum_{j=0}^{n-1} I_{i,j} * w_{i,j} \tag{1}$$

The $x_{i,j}$ represents the input feature data, and $F_{out}$ represents the output feature and $w_{i,j}$ stands for the convolution kernel, while $n \times n$ denotes the size. After transferring by original multiple dimensions LUTs, the retrieval process can be represented as:

$$F_{out} = LUT[x_{(0,0)}, \ldots, x_{(n-1,n-1)}] \tag{2}$$

In DwLUT, we utilize multiple $1 \times 1$ convolution kernels to build depthwise convolutions. As shown in Fig 3, we decompose the convolution kernel into $n^2$ of size $C_{out} \times 1 \times 1 \times 1$. Mapping by 1D LUTs, the original LUT retrieval process can be reconstructed as:

$$\hat{F}_{out} = \frac{1}{n^2} \sum_{i=0}^{n-1} \sum_{j=0}^{n-1} LUT_{(i,j)}[x_{(i,j)}] \tag{3}$$

After decomposing and mapping, the storage consumption reduces from $256^{n^2} \times C_{out} \times 8$bit to $256 \times n \times n \times C_{out} \times 8$bit. Meanwhile, the output of the multiple 1D LUTs is averaged to achieve an approximation of the standard LUT inference.

The pointwise convolution applies a $1 \times 1$ convolution to combine the outputs the depthwise convolution [36]. The operation of pointwise convolution with a kernel size of $C_{out} \times C_{in} \times 1 \times 1$ is defined as:

$$F_{out} = \sum_{c=0}^{C_{in}} \sum_{t=0}^{C_{out}} x_c \cdot w_{c,t} \tag{4}$$

In this context, $x$ represents the input data and $w$ represents weights. It still faces the problem of exponential size increase in the LUT mapping process due to multiple channels input simultaneously. Then we decouple the input channels in pointwise convolution. Subsequently, it is decoupled into $C_{in}$ number of $C_{out} \times 1 \times 1 \times 1$ convolution kernel. The storage space required for the pointwise

convolution, is reduced from $(256^{C_{in}}) \times C_{out} \times 8\text{bit}$ to $256 \times C_{in} \times C_{out} \times 8\text{bit}$. The final output channel is set to $C_{out}$. At this point, the original cross-channel per-pixel LUT retrieval operation is reconstructed as:

$$\hat{F}_{out} = \frac{1}{C_{in}^2} \sum_{c=0}^{C_{in}} LUT_c[x_c] \tag{5}$$

As depicted in Eq 5, the quantity of LUTs is denoted as $C_{in}$. Additionally, the number of output channels for each LUT retrieval is represented as $C_{out}$. According to Eq 4 and Eq 5, the PwBlock in Fig 2(b) consists of several parallel branches with the number of channels. Each branch includes 3 pointwise convolutions followed by ReLU except for the last layer. The channels in the pointwise convolution is set to 64 and that of the last layer is set to 16.

### 3.3 Dynamic Discretization Mechanism

In previous methods, in order to further reduce the volume requirement of LUT, it is common to sample the indexing entries. Related algorithms [15, 14, 37] use interpolation methods to restore accuracy after table retrieval. However interpolating in large RF and multiple channels will inevitably result in a significant amount of computation. To address this issue, we proposes the dynamic discretization mechanism (DDM). It adaptively explores the parameterized clipping level of quantization range for each channel based on the gradients with Straight-Through Estimator (STE)[38, 39], which significantly optimize the equilibrium of accuracy and LUT size.

As shown in Fig 2, an $8\text{bit}$ input data is decomposed into MSBs and LSBs, then obtain output results through their respective branches and add them. It should be noted that the 8bit data stored in the LUT is defined in the domain of $-128$ to $127$, while the data range of MSBs data is $[-m, m-1]$, and the data range of LSBs data is $[0, k]$. Therefore, in Eq 3 mapping depthwise convolution to LUTs for the MSBs branch, the LUTs size can be obtained by:

$$Size(MSB) = (2 \times m) \times n^2 \times r^2 \tag{6}$$

and the size of LUT in LSBs branch:

$$Size(LSB) = (k + 1) \times n^2 \times r^2 \tag{7}$$

This operation reduces the index range of the LUT, realizes the discretization of the LUT and reduces the size.

The index range is $2^4$ due to the activation data range are equal to LUTs indexes range in the mapping process. Therefore, inspired by activation quantization [18, 19], we use the learnable clipping parameter $\alpha$ to constrain the bit width of features. Specifically, $\alpha_{my}$ and $\alpha_{ly}$ are used to constrain the MSBs and LSBs of the layer $y$, respectively, to control the size of the LUT. For example, using $F$ to denote feature data, this operation can be calculated as:

$$F_q = round(F * \alpha_y) \tag{8}$$

The LUT size is written as :

$$Size = (max(F_q) - min(F_q)) \times n^2 \times r^2 \tag{9}$$

In Eq 9, $n^2$ denotes the input entries and $r^2$ denotes the output entries respectively. In order to reduce the index number of LUTs, we initialize $\alpha$ to $0.8$ and then apply L2-regularization for $\alpha$ with the $0.1\times$ regularization parameter $\lambda$ used to compress the activation precision scale in each layer.

## 4 Experiments

### 4.1 Implementation Details

**Training Settings** To demonstrate the effectiveness of our framework, we evaluate TinyLUT on multiple datasets for SISR. The CNN model of TinyLUT is trained in an end-to-end manner. We use DIV2K [40] dataset which has been widely applied in image processing tasks. TinyLUT model is trained for 200000 iterations with Pytorch [41] on Nvidia 3090 GPU. We employed the Adam optimizer [42], where $\beta_1 = 0.9$, $\beta_2 = 0.999$. The learning rate is set at $5 \times 10^{-3}$ and was dynamically adjusted using a cosine annealing mechanism [43]. We randomly cropped degraded data into $48 \times 48$

Table 2: Quantitative comparisons on 5 standard SISR test sets for an upscaling factor of 4. The size of TinyLUTs are smaller than other LUT methods and achieves better PSNR and SSIM average values with a good margin. ∗: The storage overhead of weight parameters in the DNN. The inference latency evaluation environments are the same as [15, 14]

|  | Method | Storage | Runtime | | Set5 | Set14 | Urban100 | BSD100 | Manga109 | Average |
|---|---|---|---|---|---|---|---|---|---|---|
|  |  |  | Xiaomi 11 | Raspberry 4B |  |  |  |  |  |  |
| LUTs | SRLUT-S [15] | 1304KB | 137ms | 247ms | 29.82/0.8478 | 27.01/0.7355 | 24.02/0.6990 | 26.53/0.6953 | 26.80/0.8380 | 26.84/0.7631 |
|  | SPLUT-L [13] | 18432KB | 265ms | 456ms | 30.52/0.8630 | 27.54/0.7520 | 24.46/0.7191 | 26.87/0.7090 | 27.70/0.8581 | 27.42/0.7802 |
|  | MuLUT-SDY-X2 [14] | 4159KB | 242ms | 403ms | 30.60/0.8653 | 27.60/0.7541 | 24.46/0.7194 | 26.86/0.7110 | 27.90/0.8633 | 27.48/0.7808 |
|  | RCLUT [34] | 1549KB | 232ms | - | 30.72/0.8677 | 27.67/0.7577 | 24.57/0.7253 | 26.95/0.7154 | 28.05/0.8655 | 27.59/0.7863 |
|  | SPF-LUT+DFC [35] | 2066KB | - | - | 31.05/0.8755 | 27.88/**0.7632** | 24.81/0.7357 | 27.08/**0.7190** | 28.58/0.8779 | 27.88/0.7943 |
|  | TinyLUT-S | **37KB** | **29ms** | **88ms** | 30.22/0.8535 | 27.33/0.7450 | 24.19/0.7066 | 26.71/0.7042 | 27.21/0.8458 | 27.13/0.7710 |
|  | TinyLUT-F | 171KB | 146ms | 387ms | **31.18/0.8771** | **28.01**/0.7630 | **24.92/0.7397** | **27.13**/0.7184 | **28.83/0.8798** | **28.01/0.7956** |
| DNN | SRCNN [12] | 228KB* | - | 27448ms | 30.48/0.8628 | 27.50/0.7513 | 24.52/0.7221 | 26.90/0.7101 | 27.10/0.8457 | 27.30/0.7784 |
|  | VDSR [24] | 2660KB* | - | 106972ms | 31.35/0.8830 | 28.02/0.7680 | 25.18/0.7540 | 27.29/0.7260 | 28.50/0.8812 | 28.07/0.8024 |
|  | FSRCNN [8] | 48KB* | 350ms | 2143ms | 30.71/0.8656 | 27.60/0.7543 | 24.61/0.7263 | 26.96/0.7129 | 27.90/0.8610 | 27.56/0.7840 |
|  | CARN-M [26] | 1648KB* | 3300ms | 17609ms | 31.82/0.8898 | 28.29/0.7747 | 25.62/0.7694 | 27.42/0.7350 | 29.85/0.8993 | 28.60/0.8136 |

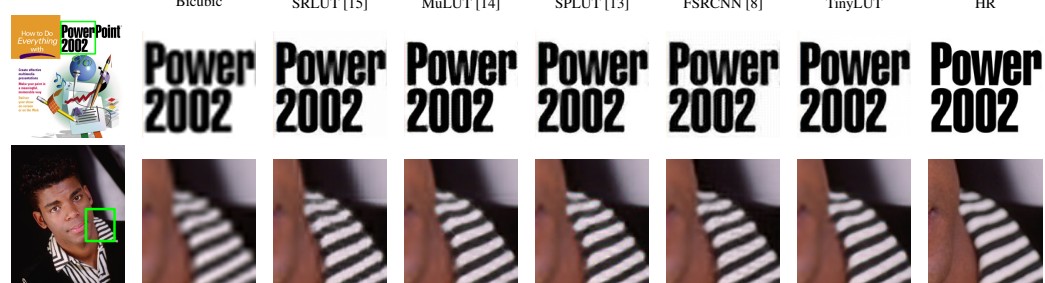

Figure 4: Qualitative comparisons of bicubic interpolation, SRLUT [15], MuLUT [14], SPLUT [13], FSRCNN [8], our TinyLUT and HR images.

patches with a batch size of 32. Data augmentation was performed through random rotations and flips. Throughout the experimental process, we employed PSNR and SSIM [44] as evaluation metrics for restoration accuracy. Besides, we measured and reported the runtime on multiple edge devices including Xiaomi 11 smartphones with a Qualcomm Snapdragon 888 CPU and Raspberry 4B.

**Evaluation Settings**   For single-image super resolution, assessed the effectiveness of our method on five widely used benchmark datasets: Set5, Set14, BSD100 [45], Urban100 [46], and Manga109 [47]. We compare our method with various SISR algorithms based on interpolation which include methods based on deep learning including FSRCNN [8], VDSR [24], RCAN [25] and CARN-M [26], and SR methods based on LUTs, SRLUT [15], MuLUT [14], RCLUT [34], SPF-LUT [35] and SPLUT [13]. In tables, the best result is highlighted in **bold**.

## 4.2   Evaluation on Image Super Resolution

**Accuracy and Storage**   The quantitative results of image super resolution are shown in Table 2. For single image super resolution task, the SSIM and PSNR values are computed at the Y-channel in the YCbCr color space. As reported in Table 2, we build a series of TinyLUT models with volume from 37KB to 171KB to fit various edge devices. Our TinyLUT-F model achieves 0.58dB PSNR increase with nearly $24\times$ storage reduction of MuLUT-SDY-X2 [14]. Even for the compact TinyLUT-S model, it only consumes $2.83\%$ memory overhead and achieves 0.4dB PSNR and 0.02 SSIM improvement compared to SRLUT-S [15]. The qualitative results in Fig 4 of MuLUT-SDY-X2 [14] and SPLUT-L [13] have severe ringing artifacts in the white area. On the contrary, our method generates more natural textures and fewer artifacts. As shown in Table 2, TinyLUT-F achieves much higher PSNR with $9\times$ and $12\times$ lower storage consumption than RCLUT [34] and SPF-LUT+DFC [35], respectively. While TinyLUT achieves restoring clearer edges and $5\times$ lower latency on Raspberry 4B compared with computation-heavy method(FSRCNN[8]).

Table 3: Quantitative comparisons on Set12 and BSD68. The size of TinyLUTs are smaller than other LUT methods and achieves better PSNR and SSIM values with a good margin. ∗: The storage overhead of weight parameters in DNN. The evaluation environments are the same as [48]

| Method | Storage | Runtime | | Set12 | | | BSD68 | | | Average |
| --- | --- | --- | --- | --- | --- | --- | --- | --- | --- | --- |
| | | Xiaomi 11 | Raspberry 4B | 15 | 25 | 50 | 15 | 25 | 50 | |
| SRLUT [15] | 82KB | **7ms** | **21ms** | 30.42 | 27.19 | 22.62 | 29.78 | 26.85 | 22.39 | 26.54 |
| MuLUT-SDY-X2 [14] | 289KB | 26ms | 44ms | 31.50 | 28.94 | 25.46 | 30.63 | 28.18 | 24.97 | 28.28 |
| MuLUT-SDYEHO-X2 [48] | 978KB | 51ms | 89ms | 31.77 | 29.18 | 25.47 | 30.89 | 28.34 | 24.96 | 28.44 |
| TinyLUT-S | **22KB** | 20ms | 27ms | 31.10 | 28.26 | 24.29 | 30.24 | 27.48 | 23.83 | 27.53 |
| TinyLUT-F | 187KB | 146ms | 254ms | **32.22** | **29.69** | **26.27** | **31.20** | **28.65** | **25.53** | **28.93** |
| DnCNN [7] | 2220KB* | 635ms | 6859ms | 32.86 | 30.44 | 27.18 | 31.73 | 29.23 | 26.23 | 29.61 |

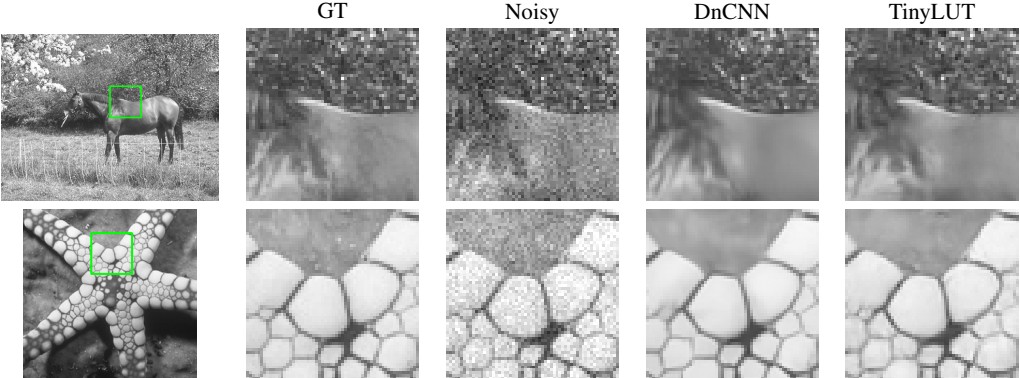

GT        Noisy        DnCNN        TinyLUT

Figure 5: Qualitative comparisons of ground truth, noise image, DnCNN [7] and our TinyLUT.

**Running Time**    Inference speed is also an important impact for edge devices deployment. In the view of Table 2, we report the runtime of LUT-base methods and other schemes. Results are obtained by averaging across 20 runs. Our approach achieves $4\times$ and $10\times$ acceleration on SISR compared to the fastest LUT-based and CNN method respectively. Meanwhile TinyLUT obtains absolute accuracy advantage compared to LUT-based methods. Overall the experiments indicate that the SMS is achieving significant storage reduction while organizing the model in a more flexible manner, thus capable of improving accuracy. TinyLUT achieves accuracy close to CNN and exhibits the enormous application potential of LUT-based method.

### 4.3    Evaluation on Image Denoising

In order to further show the capacity of the proposed TinyLUT framework, we remove the pixelshuffle and set the out channels of the last PwLUT to 1. Meanwhile, we adopt the residual learning formulation such as DnCNN [7] to accelerate model training. For additive white Gaussian noise (AWGN), we use Set12 and BSD68 benchmark datasets with noise level 15, 25 and 50. We compare our method with classical DNN method [7] and LUT methods [15, 48].

The quantitative results of image denoising is shown in Table 3. Compare with LUT-based methods, our method significantly improves PSNR with the minimal storage in image denoising. Meanwhile, TinyLUT-S achieves the fastest inference speed with only 22KB storage overhead compare with DNN based denoising schemes. The denoising results in Fig 5 illustrate that our TinyLUT is able to obtain similar visual quality to DnCNN [7]. The quantitative and qualitative experiments prove the effectiveness of storage reduction and inference acceleration of our approach.

### 4.4    Evaluation on Image Deblocking

In this subsection, image deblocking is used to further assess the generality of TinyLUT in image restoration tasks. Table 4 reports the comparison results of PSNR-B for LUT-based methods on Classic5[49] and LIVE1[50]. We also refer to the classical methods and DNN algorithms, including SA-DCT[51] and ARCNN[52]. TinyLUT-F achieves comparable PSNR-B results to the DNN method

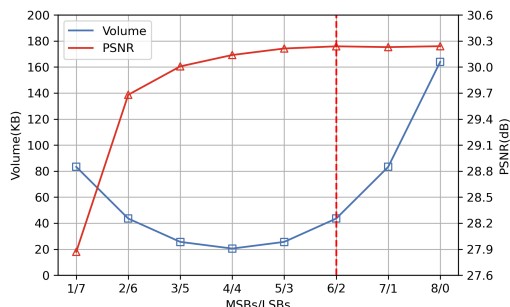

Figure 6: Illustration of the storage overhead and PSNR result for different combinations of MSBs and LSBs in TinyLUT-S for single image super resolution task.

Figure 7: Illustration of the input entities precision range clipping by parameter $\alpha_{MSB}$ when mapping process in TinyLUT-S-$\alpha$ in SISR. TinyLUT-S-E indicates the input entities range of PwLUT as a statistical result.

and better than other LUT-based methods. The results indicate the generality of our TinyLUT to deblocking.

Table 4: The quantitative comparisons of PSNR-B on standard benchmark datasets for image deblocking under a quality factor of 10.

|  | Classical | | DNN | LUT-based | | | | |
|---|---|---|---|---|---|---|---|---|
|  | JPEG | SA-DCT[51] | ARCNN[52] | SRLUT[15] | MuLUT[14] | SPF-LUT[35] | SPF-LUT+DFC[35] | TinyLUT-F |
| Storage | - | - | 415KB | **81KB** | 489KB | 3017KB | 595KB | 187KB |
| Classic5 | 25.21 | 28.15 | 28.76 | 27.58 | 28.29 | 28.63 | 28.62 | **28.74** |
| LIVE 1 | 25.33 | 28.01 | 28.77 | 27.69 | 28.39 | 28.62 | 28.61 | **28.67** |

## 4.5 Ablation Studies

We perform several ablation experiments for single image super resolution to demonstrate that our contributions in TinyLUT provides quality improvements.

**The effectiveness of SMS** We conduct an experiment with combinations of SMS and TinyLUT-S. In Table 5, SMS denotes the TinyLUT-S includes SMS without DDM when constructed the model with 8bit data values. Original denotes the TinyLUT-S without SMS and evaluates using the quantized CNN model due to the excessive storage overhead of its LUT model. The accuracy of SMS is competitive with corresponding mapped neural network with 8bit data in SISR. In particular, SMS scheme achieves significant accuracy improvement compared to uniformly sampled method [15] with sampling interval size of $2^4$, while yielding over **7×** storage reduction. This demonstrates the effectiveness of storage reduction using SMS with minimal loss of accuracy.

Table 5: Impact of SMS. Ablation studies on TinyLUT-S for $4\times$ SISR.

| Model | Method | Set5 | Set14 | Urban100 | BSD100 | Manga109 | LUT Size |
|---|---|---|---|---|---|---|---|
| | Original | 30.35 | 27.42 | 26.77 | 24.31 | 27.41 | $9.7 \times 10^{24}$PB |
| TinyLUT-S | Uniformly Sampled [15] | 29.82 | 27.01 | 26.53 | 24.02 | 26.80 | 1.274MB |
| | SMS | 30.24 | 27.33 | 26.72 | 24.19 | 27.23 | **164KB** |

**The effectiveness of DDM** As shown in Fig 6, the PSNR results of super resolution and image denoising are approximately equal to the global maximum at 6 MSBs and 2 LSBs (6M2L). Meanwhile the storage overhead is much smaller than the combination of 5 MSBs and 3 LSBs. Therefore we set the activation precision for the two branches of TinyLUT to 6 MSBs and 2 LSBs respectively.

As shown in Table 6, compared with the full LUT with $2^8$ entries for each input pixel, DDM reduces TinyLUT storage overhead while ensuring accuracy in super resolution. The results prove that the

Table 6: Effects of DDM. Ablation studies for DDM with $4\times$ SISR.

| Model | Method | Set5 | Set14 | Urban100 | BSD100 | Manga109 | LUT Size |
|---|---|---|---|---|---|---|---|
| TinyLUT-S | Full LUT | 30.24 | 27.33 | 26.72 | 24.19 | 27.23 | 164KB |
| | DDM | 30.22 | 27.33 | 26.71 | 24.19 | 27.21 | **37**KB |

LUT sampling method based on DDM has higher compression ratio of storage than the 8bit full sampling LUT, and achieves a better trade-off between storage and accuracy.

As shown in Fig 7, the precision range of the MSBs activation data is adjusted more adaptively by clipping parameter $\alpha$ in SISR, which reduces the possible value range of the input entities of PwLUT in TinyLUT-S, and then reduces the size in MSBs branch. As a comparison, the marked red line in the figure is the $s$ value when the input entities range is fixed to 6bit during mapping in the MSBs of PwLUT. It can be observed from the figure that compressed by the DDM, not only the LUT size is reduced, but also all possible values of input entities are covered in the mapping process. Compared with the 43.6KB of TinyLUT-S-6M2L with a fixed combination of 6 MSBs and 2 LSBs, the size of TinyLUT-S-E is reduced to 36.6KB via the adjustment of DDM, achieving a reduction of $15\%$. Therefore, DDM yields additional storage reduction compared to decomposing data into fixed MSBs and LSBs [13]. More intuitively, the interpretability of DDM is also illustrated in Fig 7.

## 5   Limitations and Future Direction

The method in this paper achieves significant reduction in storage with high compatibility when mapping convolutional neural network. However, there are difficulties in mapping other models such as Mamba [53] and Transformer [54, 55] using TinyLUT. This can be further explored the unified mapping approach for other model. Exploring additional image restoration tasks on edge devices using LUTs would also contribute to the community.

## 6   Conclusion

In this paper, we analyze previous successful LUT-based deep learning approaches and summarize the key problem of storage explosion, which limits the further popularization of LUT in image restoration on edge devices. To address the storage explosion, we propose the separable mapping strategy (SMS) and dynamic discretization mechanism(DDM) to decompose the kernel and activation, respectively. In particular, we design the TinyLUT framework based on SMS and DDM. By seamlessly integrating these innovations, TinyLUT-F sets a new record for SISR by achieving over 31dB PSNR on the Set5 dataset at just 171KB LUT storage. Overall, extensive experiments across seven benchmark datasets and two classic image restoration tasks demonstrate the effectiveness and efficiency of TinyLUT on resource-constrained devices.

## 7   Acknowledgement

This work was supported in part by the National Science and Technology Innovation 2030-Major Projects under Grant 2021ZD0114400, in part by Young Scientists Fund of the National Natural Science Foundation of China 62304162, and in part by China Postdoctoral Science Foundation under Grant 2024M762532, and in part by Postdoctoral Fellowship Program of CPSF under Grant GZC20241313, and in part by Shaanxi Provincial Natural Science Foundation for Basic Research Program 2024JC-YBMS-794, and in part by Fundamental Research Funds for the Central Universities under Grant XJSJ24090. Dr. Lin Gu was supported by JST Moonshot R&D Grant Number JPMJMS2011 Japan.

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
