# OpenReview forum: "TinyLUT: Tiny Look-Up Table for Efficient Image Restoration at the Edge"
_NeurIPS.cc/2024/Conference — NeurIPS 2024 poster_

### Official Review · Reviewer_MFFN · 2024-06-25

**Soundness:** 3
**Presentation:** 3
**Contribution:** 3
**Rating:** 7
**Confidence:** 5

**Summary:**

To address the storage explosion challenge of LUT, this paper proposes a separable mapping strategy (SMS) and a dynamic discretization mechanism (DDM) to decompose the kernel and activation to reduce the storage consumption. Specifically, the SMS decomposes the convolution into independent sub-operations to reduce the input entries, consequently reducing the dimension of LUT. Additionally, the DDM explicitly compresses the quatization scales with learnable clipping parameters to decrease LUT scales.

**Strengths:**

1. This paper analyzes the storage explosion challenge of LUT and provides a solution by decomposing the convolution kernel and compressing the quantization scale.
2. The proposed image restoration method is efficient and effective for edge devices, as demonstrated in the experiments.

**Weaknesses:**

1. Although the separable mapping strategy (SMS) could significantly reduce the storage, it may lead to information loss and performance drop. As shown in Figure 3, the SMS obtains the final result using individually mapped indexes and neglects the relationship between the local indexes, which is important for convolutions.
2. The motivations and benefits of activation decomposition in the DDM are unclear. Moreover, this paper does not sufficiently discuss the difference between DDM and former activation decomposition used in SPLUT.
3. The discussion of LUT-based methods in the related works is insufficient. The key ideas and limitations of these methods should be elaborated.

**Questions:**

1. To my understanding, the proposed strategy can not be applied to transformer-based model like SwinIR. Why is SwinIR introduced in the related works, and why is it categorized under CNN methods?
2. Different from previous methods, the proposed method is not contrained by the storage. Can the proposed method achieve better performance by increasing the RF and channels?
3. Why is the denoising experiment placed after the ablation experiments? The reviewer suggests combining it with the SR evaluation.

**Limitations:**

The limitations have been discussed in the paper.

---

> ### Author Rebuttal · Authors · 2024-08-07
>
> **Response to Weakness 1** We thank the reviewers for their comprehensive review. It must be clarified that the proposed SMS strategy will not result in information loss or performance degradation due to neglecting the relationships between local indexes.  As for the original depthwise convolution operation, it can be obtained as follows:
> \begin{equation}
> 	F_{out} = \sum_{i=0}^{n-1}\sum_{j=0}^{n-1} {I_{i,j}*w_{i,j}}
> \end{equation}
> The reconstruction of local index relationships is achieved through a summation process. This process involves the multiplication of each pixel within the receptive field with its corresponding weight. The resulting products are then aggregated to form the final output.
>
> After the CNN model training, we stored the results of each pixel within the receptive field multiplied with the trained weights in the corresponding LUT.
> \begin{equation}
> 	LUT_{(i,j)}[I_{(i,j)}] = I_{i,j}*w_{i,j}
> \end{equation}
> Similarly, after obtaining individual input-corresponding results through separately mapped indices in SMS, the final result is constructed via summation of all outputs. The relationships between local indices are reconstructed in this process. This process is formally expressed in the next equation of our manuscript:
> \begin{equation}
> 	\hat F_{out} = \frac{1}{n^{2}} \sum_{i=0}^{n-1}\sum_{j=0}^{n-1} LUT_{(i,j)}[x_{(i,j)}]
> \end{equation}
> Following the discrete retrieval in SMS, a summation of individual results is necessitated to reconstruct the relationships among local indices. To prevent misinterpretation, we will refine Figure 3 to explicitly illustrate this crucial step in the process.
>
> **Response to Weakness 2** We appreciate the reviewer's insightful question. Our motivation for DDM stems from the limitations of the previous SPLUT approach, which was constrained to a 4 MSBs and 4 LSBs symmetric decomposition scheme for 8-bit inputs.  SPLUT uses a fixed quantization scale for each layer. These constraints inherently limited model performance.
>
> Our DDM proposed an innovative asymmetric activation decomposition strategy and adaptively finds the right quantization scale for each layer, thus significantly boosting the inference accuracy and achieving a better balance between accuracy and storage.
> Notably, the DDM strategy is not exclusive to our method. It can be applied to other LUT-based methods to further reduce the storage consumption.
> ﻿
>
>
>
> **Response to Weakness 3** The reviewer's reminder is very important and insightful. In response, we will augment the related work section with an explication of the methodologies presented in [1] and [2], the key ideas and limitations are also will be elaborated. In addition to summarizing the key ideas and limitations of SRLUT, SPLUT and MULUT, we will also summarize RCLUT's proposal of the plugin module to increase the RF of the model with slightly increasing storage overhead. SPFLUT[1] proposed a LUT compression framework to balance the performance improvement and storage growth. The incorporation of these references will enhance the comprehensiveness of our literature review.
>
> **Response to Question 1** We extend our gratitude for the reviewer's meticulous examination. The inclusion of SwinIR in the related work section was motivated by its significant performance. However, upon careful consideration of the reviewer's astute observation, SwinIR does not align with the CNN-centric theme of our work, and the inappropriate description is removed from the manuscript.
>
> Furthermore, we restructure the categorization of methods in the related work section. Specifically, we will supplement [3] and [4] under the CNN methods subsection to provide a more comprehensive overview of CNN approaches.
> We reiterate our appreciation for the reviewer's thorough and insightful feedback, which has significantly contributed to enhancing the coherence and relevance of our manuscript.
>
> **Response to Question 2** We appreciate the reviewer's suggestion for improvement. We validated the suggestions of the reviewer in the 4x SR task. As shown in Table 1, the improvement resulting from using a 5x5 RF and 32 feature channels shows an increase in accuracy with only a minor linear growth in storage overhead compared to the baseline.
>
> We extend our sincere gratitude to the reviewers for their insightful suggestions, which have illuminated additional avenues for exploration in our method.
>
> **Table 1 Quantitative comparisons on 4x SR**
> |                     | Storage | Set5  | BSD100 | Manga109 |
> |---------------------|---------|-------|--------|----------|
> | TinyLUT-S           |    37KB     | 30.22 | 26.71  | 27.21    |
> | TinyLUT-5x5 RF      |   60KB      | 30.30 | 26.75  | 27.24    |
> | TinyLUT-32 channels |    136KB     | 30.35 | 26.77  | 27.39    |
>
> **Response to Question 3** The reviewer's suggestion is very important and insightful. We will revise the structure of the experimental section and combine the denoising experiments with the SR evaluation. This reorganization will significantly improve the clarity and flow of our research presentation.
>
> [1] Li et al. "Look-Up Table Compression for Efficient Image Restoration", CVPR2024
>
> [2] Liu et al. "Reconstructed Convolution Module Based Look-Up Tables for Efficient Image Super-Resolution", ICCV2023
>
> [3] Lai et al. "Fast and Accurate Image Super-Resolution with Deep Laplacian Pyramid Networks", TPAMI 2019
>
> [4] Tassano et al. "FastDVDnet: Towards Real-Time Deep Video Denoising Without Flow Estimation", CVPR2018

---

> > ### Comment · Reviewer_MFFN · 2024-08-13
> > **Post Rebuttal Comments**
> >
> > Thank you for the careful explanation. After reviewing the rebuttal and considering the feedback from other reviewers, my concerns have been addressed. I have raised my score to Accept, as this work effectively tackles the storage explosion of LUTs and provides a valuable image restoration method for edge devices.

---

> > > ### Author Response · Authors · 2024-08-14
> > > **Thank you for engaging in the discussion!**
> > >
> > > We would like to thank the reviewer for engaging in the discussion and increasing the score. We will ensure that all edits mentioned in the rebuttal are incorporated when revising our paper. Thanks again for your participation in the discussion!

---

### Official Review · Reviewer_2kHn · 2024-07-11

**Soundness:** 3
**Presentation:** 3
**Contribution:** 2
**Rating:** 5
**Confidence:** 5

**Summary:**

This paper introduced a separable mapping strategy that solves the storage issue of LUT-based methods. In addition, a dynamic discretization mechanism is designed to decompose the activation and compression quantization scales. Experimental results show the potential of this work for image restoration tasks.

**Strengths:**

1. The paper was well written and organized.

2. The performance improvement of SR task is remarkable.

3. The proposed method is valuable for advancing image restoration on edge devices.

**Weaknesses:**

1. There is a lack of comparison with the state-of-the-art methods [1, 2], especially the literature [1] which also addresses the storage problem of LUT-based methods.

2. This paper claims to work on the image restoration problem, however only two restoration tasks were included in the experiments. Validation on more image restoration tasks is needed.

3. The proposed method does not perform well enough on the image denoising task. With similar inference efficiency, the performance of the proposed method lags far behind MULUT-SDY-X2 even more than 1 dB. I am concerned about the effectiveness of this work on a wider range of image restoration tasks.


> 1. Look-Up Table Compression for Efficient Image Restoration. CVPR 24.

>2. Reconstructed Convolution Module Based Look-Up Tables for Efficient Image Super-Resolution. ICCV 23.

**Questions:**

Please see Weaknesses.

**Limitations:**

Limitations were included.

---

> ### Author Rebuttal · Authors · 2024-08-07
>
> **Response to W1** The reviewer's reminder is very important and constructive. To address the reviewer's concern, we conduct a comparative study with the abovementioned two methods[1,2]. The comparison indicates that our TinyLUT is more effective than the latest storage saving LUT methods. Literature [1] proposed a LUT compression framework named DFC for efficient image restoration with a different technological route from our proposed TinyLUT. The DFC preserves diagonal LQHQ pairs to maintain representation capacity, while non-diagonal pairs are aggressively subsampled to save storage. Method [2] proposed a reconstructed convolution to decouple the calculation of spatial and channel-wise features for maintaining large RF with less LUT size.
> As shown in Table 1, TinyLUT-F achieves much higher PSNR with 9× and 12× lower storage consumption than RCLUT [2] and SPFLUT+DFC [1], respectively.
>
> **Table 1 Quantitative comparisons on 4x SR**
> |                | storage  | Set5  | Set14 | BSD100 | Manga109 | Urban100 |
> |----------------|----------|-------|-------|--------|----------|----------|
> | RCLUT [2]      | 1.513MB  | 30.72 | 27.67 | 26.95  | 28.05    | 24.57    |
> | SPFLUT [1]        | 17.284MB | 31.11 | 27.92 | 27.10  | 28.68    | 24.87    |
> | SPFLUT+DFC [1] | 2.018MB  | 31.05 | 27.88 | 27.08  | 28.58    | 24.81    |
> | TinyLUT-F      | 171KB    | 31.18 | 28.01 | 27.13  | 28.83    | 24.92    |
>
> In addition, we compare SPFLUT+DFC [1] and TinyLUT in image desnoing with noise level 15. Note that RCLUT [2] did not test on image denoising task. As shown in Table 2, TinyLUT still yields about 0.2dB higher PSNR with 3× storage reduction compared to SPFLUT+DFC. The results further demonstrate that TinyLUT exhibits notable advantages in other image restoration tasks. These experiments corroborate the effectiveness of the proposed method, as noted by reviewers (G5Ti, MFFN).
>
> **Table 2 Quantitative comparisons on image denoise**
> |            | storage | Set12 | BSD68 |
> |------------|---------|-------|-------|
> | SPFLUT     | 3017KB  | 32.11 | 31.17 |
> | SPFLUT+DFC | 595KB   | 32.01 | 31.09 |
> | TinyLUT-F  | 187KB   | 32.22 | 31.20 |
>
> **Response to W2** Thanks for the reviewer's insightful suggestion. As claimed in [3,4], image denoising and SR are classical yet still active topics in low-level vision since they are indispensable steps in many practical applications. Most of the image restoration models[3,4] adopted denoising and super-resolution as representative tasks. Upon the reviewer's insightful comment, we fully agree that the limitations of focusing solely on SR and image denoising cannot comprehensively demonstrate the generality in image restoration.
>
> To address this concern, we incorporate the deblocking task widely employed in recent literature [1] into our experiment. The results of this experiment demonstrate the value of our method for advancing image restoration on edge devices. As shown in Table 3, TinyLUT-F still achieves remarkably higher PSNR-B than other LUT methods with much lower storage consumption.
>
> **Table 3 Quantitative comparisons on image deblock**
> |            | storage | Classic5 | LIVE1 |
> |------------|---------|----------|-------|
> | SRLUT      | 81KB    | 27.58    | 27.69 |
> | MULUT      | 489KB   | 28.29    | 28.39 |
> | SPFLUT     | 3017KB  | 28.63    | 28.62 |
> | SPFLUT_DFC | 595KB   | 28.62    | 28.61 |
> | TinyLUT-F  | 187KB   | 28.74    | 28.67 |
>
> **Response to W3** The reviewer's reminder is very important. In the image denoising task, our TinyLUT-S does lag far behind MULUT-SDY-X2 in accuracy, which is results from the significant reduction in storage for deploying on edge devices with very limited resource budgets, especially memory and storage [5,6]. As shown in Table 4, our proposed TinyLUT-S occupies merely 7.6% of the storage required by MULUT-SDY-X2 that is the most lightweight version of MULUT.
>
> As for the inference efficiency, inspired by comments from reviewers G5Ti and 2kHn, we found that still some operations can be merged to reduce inference latency, such as residual addition between blocks. Meanwhile, we have introduced pthread and omp to accelerate inference in a multi-threaded parallel manner. As a result, the inference latency of TinyLUT-S is reduced to 27ms on Raspberry Pi 4B, yielding a real-time inference efficiency comparable to the current lightest SRLUT with about 1dB PSNR promotion and 4× lower storage. Hence, our TinyLUT achieves a better trade-off between accuracy and latency, as noted by reviewer G5Ti.
>
> **Table 4 Quantitative comparisons on denoise and latency**
> |                 | Runtime | Storage | Set12 | BSD68 | Set12 | BSD68 | Set12 | BSD68 | Average |
> |-----------------|---------|---------|-------|-------|-------|-------|-------|-------|---------|
> | SRLUT           | 21ms    | 82KB    | 30.42 | 29.78 | 27.19 | 26.85 | 22.62 | 22.39 | 26.54   |
> | MULUT-SDY-X2    | 44ms    | 289KB   | 31.50 | 30.63 | 28.94 | 28.18 | 25.46 | 24.97 | 28.28   |
> | MULUT-SDYEHO-X2 | 89ms    | 978KB   | 31.77 | 30.89 | 29.18 | 28.34 | 25.47 | 24.96 | 28.44   |
> | TinyLUT-S       | **27ms**    | 22KB    | 31.10 | 30.24 | 28.26 | 27.48 | 24.29 | 23.83 | 27.53   |
>
> In addition, we compare our method and other LUT-based methods in image deblocking task under a quality factor of 10 in Table 3. The results demonstrate the effectiveness of our work on other image restoration tasks.
>
> [1] Li et al. "Look-Up Table Compression for Efficient Image Restoration", CVPR2024
>
> [2] Liu et al. "Reconstructed Convolution Module Based Look-Up Tables for Efficient Image Super-Resolution", ICCV2023
>
> [3] Zhang et al. "Beyond a gaussian denoiser: Residual learning of deep CNN for image denoising", TIP 2017
>
> [4]  Kim et al. "Accurate Image Super-Resolution Using Very Deep Convolutional Networks", CVPR 2016
>
> [5] Lin et al. "MCUNet: Tiny Deep Learning on IoT Devices",  NIPS 2020
>
> [6] Lin et al. "MCUNetV2: Memory-Efficient Patch-based Inference for Tiny Deep Learning", NIPS2021

---

> > ### Comment · Reviewer_2kHn · 2024-08-13
> > **Reply**
> >
> > Thanks to your response, some of my concerns were addressed. The authors' response addresses the ethical issues well. However, I remain concerned about the performance of this work on a wider range of image restoration tasks. The performance of the proposed method lags too far behind on the denoising task, even though it is more efficient. It is difficult to argue that the method achieves a better trade-off between performance and efficiency, especially with respect to the MULUT-SDY-X2 method. Therefore, I would like to keep the previous rating.

---

> > > ### Author Response · Authors · 2024-08-13
> > >
> > > Thanks for the reviewer's reply. As demonstrated by the reviewer, the accuracy of TinyLUT-S in denoising is indeed lower than MULUT-SDY-X2, although TinyLUT-S is more efficient in storage and inference latency. However, it should also be noted that the denoising accuracy on the larger TinyLUT-F version is significantly higher than all existing LUT models, with an average increase of 0.6dB compared to MULUT-SDY-X2 and 35% storage reduction. Meanwhile, TinyLUT-F achieves a 0.49dB accuracy increase over MULUT-SDYEHO-X2 with only 19% storage consumption. While the accuracy of methods such as MULUT reaches its upper limit due to storage limitations. This indicates that our method addresses the storage explosion challenge of LUT-based methods, which is also the most important contribution of this paper and achieves a better balance between storage and accuracy.
> > >
> > > Following the reviewer's insightful suggestions, we used multithreading and residual merging operations to accelerate the inference latency from 383ms to 254ms, and we noticed that TinyLUT-F still has potential for optimization in speed, such as referencing the proven efficiency enhancing LayerMerge [1] technique and other merging methods [2,3,4]. Thanks to the reviewer for the suggestions and questions regarding inference efficiency. This has allowed us to identify the shortcomings of TinyLUT in balancing inference efficiency and accuracy, which will also be an important direction for optimizing TinyLUT models in the future.
> > >
> > > [1] Kim et al. “LayerMerge: Neural Network Depth Compression through Layer Pruning and Merging”, ICML2024
> > >
> > > [2] Kim et al. “Efficient latency-aware cnn depth compression via two-stage dynamic programming”, ICML2023
> > >
> > > [3] Dror et al. “Layer folding: Neural network depth reduction using activation linearization”, BMVC2022
> > >
> > > [4] Fu et al. “Depthshrinker: A new compression paradigm towards boosting real-hardware efficiency of compact neural networks”, ICML2022

---

> > > ### Author Response · Authors · 2024-08-14
> > > **Thank you for engaging in the discussion!**
> > >
> > > We would like to thank the reviewer for engaging in the discussion. We will ensure that all edits mentioned in the rebuttal are incorporated when revising our paper. Regarding the inference efficiency you mentioned, it will be one of the directions for our future research. Thanks again for your participation in the discussion!

---

### Official Review · Reviewer_G5Ti · 2024-07-12

**Soundness:** 3
**Presentation:** 3
**Contribution:** 3
**Rating:** 6
**Confidence:** 3

**Summary:**

The paper presents TinyLUT, a method that significantly reduces LUT-based image restoration storage requirements for edge devices through separable mapping strategy and dynamic discretization mechanism, achieving competitive accuracy with over 5 times faster inference.

**Strengths:**

1. The proposed separable mapping strategy (SMS) and dynamic discretization mechanism (DDM) are efficient and effective for restoration tasks.
2. The proposed TinyLUT achieves significant reduction in memory storage and also gains better accuracy-latency tradeoff over other methods.
3. The paper demonstrates the great potential of LUT-based methods.

**Weaknesses:**

1. The proposed SMS and DDM are used to built TinyLUT model, so how can these two modules help to improve other models? That is, it would be better if you can validate their generalization as general compression modules.
2. Should compare to more recent works, like [a].

[a]. Look-Up Table Compression for Efficient Image Restoration. Li. CVPR 2024

**Questions:**

1. How do the models deploy in edge devices? Which computing engine or accelerating library is used? I think it is helpful for readers to understand the efficiency of the LUT-based methods.

---

> ### Author Rebuttal · Authors · 2024-08-07
>
> **Response to Weakness 1** The reviewer's comment is very insightful. The proposed SMS and DDM methods are applicable to other LUT-based models such as SRLUT[1], SPLUT[2] and MULUT[3].
>
> Our proposed SMS enables decomposition of the input from RF and channels of these models for fewer LUT storage overhead. For example, the storage of SRLUT-S can be reduced from 1.27MB to 16KB by SMS. Moreover, the addition of DDM will further reduce the storage from 16KB to about 4.25KB to adapt to the deployment on edge devices. Due to time constraints during the rebuttal period, comprehensive validation of this aspect will be presented in a subsequent appendix.
>
> **Response to Weakness 2** The reviewer's suggestion is very valuable. To demonstrate the advantages of our method, we have conducted comparisons on image super-resolution and image denoising tasks with method [4] in Table 1. As described in your 'Strengths' section, these experimental outcomes further substantiate the considerable potential of LUT-based approaches. The method [4], being another LUT compression framework, was published after the submission of our manuscript. Method [4] based on DFC strategy preserves diagonal pairs to maintain the representation capacity, while non-diagonal pairs are aggressively subsampled to save storage.  As can be seen, TinyLUT-F achieves the highest PSNR with about 12x storage reduction compared to SPFLUT+DFC.
> Since DFC and TinyLUT are different technological routes, we can combine them to achieve greater storage reduction.
>
> **Table 1 Quantitative comparisons on 4x SR task**
> |            | storage  | Set5  | Set14 | BSD100 | Manga109 | Urban100 |
> |------------|----------|-------|-------|--------|----------|----------|
> | SPFLUT     | 17.284MB | 31.11 | 27.92 | 27.10  | 28.68    | 24.87    |
> | SPFLUT+DFC | 2.018MB  | 31.05 | 27.88 | 27.08  | 28.58    | 24.81    |
> | TinyLUT-F  | 171KB    | 31.18 | 28.01 | 27.13  | 28.83    | 24.92    |
>
> In addition, we compare SPFLUT+DFC and our TinyLUT in the image denoising task with noise level 15 on Set12 and BSD68. As shown in Table 2, TinyLUT still yields about 0.2dB higher PSNR with 3× storage reduction compared to SPFLUT+DFC. The results further demonstrate that TinyLUT exhibits notable advantages in other image restoration tasks. These experiments corroborate the effectiveness of the proposed method, as noted by reviewers (G5Ti, MFFN).
>
> **Table 2 Quantitative comparisons on image denoise**
> |            | storage | Set12 | BSD68 |
> |------------|---------|-------|-------|
> | SPFLUT     | 3017KB  | 32.11 | 31.17 |
> | SPFLUT+DFC | 595KB   | 32.01 | 31.09 |
> | TinyLUT-F  | 187KB   | 32.22 | 31.20 |
>
> **Response to Question 1** The inference speed evaluation was conducted by deploying TinyLUT on two distinct platforms: Xiaomi 11 smartphone and Raspberry Pi 4B.
> For the Xiaomi 11 platform, the TinyLUT was implemented in C++ and  invoked by Java Native Interface. Multi-thread parallel acceleration for look-up table was achieved through the Stream API.
> For the Raspberry Pi 4B implementation, the TinyLUT model was also coded in C++. We employed the "pthread" library and "omp" library for multi-thread look-up table acceleration .
> The source code will be made publicly available on GitHub to help the readers to understand the efficiency of the LUT-based methods.
>
> [1] Jo et al. "Practical Single-Image Super-Resolution Using Look-Up Table", CVPR2021
>
> [2] Ma et al. "Learning Series-Parallel Lookup Tables for Efficient Image Super-Resolution", ECCV2022
>
> [3] Li et al. "MuLUT: Cooperating Multiple Look-Up Tables for Efficient Image Super-Resolution", ECCV2022
>
> [4] Li et al. "Look-Up Table Compression for Efficient Image Restoration", CVPR2024

---

> ### Author Response · Authors · 2024-08-14
> **Thank you for engaging in the discussion!**
>
> We greatly appreciate your time and dedication to providing us with your valuable feedback. We hope we have addressed the concerns, but if there is anything else that needs clarification or further discussion, please do not hesitate to let us know. We will ensure that all edits mentioned in the rebuttal are incorporated when revising our paper. Thanks again for your participation in the discussion!

---

### Official Review · Reviewer_JnS6 · 2024-07-13

**Soundness:** 3
**Presentation:** 2
**Contribution:** 2
**Rating:** 3
**Confidence:** 4

**Summary:**

This paper proposes to reduce the size of LUT for image restoration and to make it applicable on edge devices. The main idea is using depthwise separable convolution to replace the vanilla convolution. When transfer to LUT, the storage is significantly reduced. Experiments show the effectiveness and efficiency of the proposed method.

**Strengths:**

The approach is reasonable and the experiment result is provable.
The evaluation is complete. The efficiency of the proposed method on edge device is proved.

**Weaknesses:**

The baseline DnCNN is somehow out-of-state. As described in Limitation, the proposed method based on depthwise separable convolution is suitable for CNN based architectures only, and the LUT-based compression on attention-based method is to be explored.

Quantitative comparisons on color image restoration tasks are needed, e.g., CBSD68. It is better to conduct experiments on more testsets.

In runtime comparison, it is necessary to compare the complexity of the comparison algorithms. I am interested in the processing time of the proposed method on high-resolution images.

The description of the method is hard to understand. The writing should be improved.

**Questions:**

See the weaknesses above.

Please follow the code of ethics in [1], Lena image in Fig 2 may violate this.

[1] https://conferences.ieeeauthorcenter.ieee.org/write-your-paper/improve-your-graphics/

**Limitations:**

Yes

---

> ### Author Rebuttal · Authors · 2024-08-06
>
> **Response to Weakness 1**  We appreciate the reviewer's meticulous work. Indeed, our proposed TinyLUT as well as existing LUT-based methods are mainly designed for CNN-based architectures that are widely deployed on resource-constrained edge devices. Primarily because the convolution operation is computationally efficient [1][2][3]. CNNs compete favorably with Transformers in terms of accuracy and scalability, and convolution remains much desired and has never faded[4][5]. In recent years, many CNN methods have still been proposed [4][5][6].
>
> Meanwhile, as highlighted by the reviewer and acknowledged in our Limitations section, LUT-based compression for attention-based methods represents a critical direction for future research. This area will be a focal point in our subsequent investigations.
>
> **Response to Weakness 2** We thank the reviewer for the constructive suggestion. To address the reviewer's concern, we have conducted supplementary experiments for denoising task under 50 noise level on more color image test sets, inculding Kodak, CBSD68 and McMaster. The experiment results further indicate the effectiveness of our proposed TinyLUT, corroborating the assertions made by reviewers MFFN and G5Ti. The experimental results are shown as follows：
>
> **Table 1 Quantitative comparisons on color image denoising with noise level 50**
> |                 | Storage | Kodak | CBSD68 | McMaster | Average |
> |-----------------|---------|-------|--------|----------|---------|
> | SRLUT           | 82KB    | 22.64 | 22.45  | 23.41    | 22.83   |
> | MULUT-SDY-X2    | 489KB   | 25.64 | 24.95  | 26.38    | 25.66   |
> | MULUT-SDYEHO-X2 | 979KB   | 25.80 | 25.07  | 26.57    | 25.81   |
> | TinyLUT-F       | 189KB   | 26.43 | 25.53  | 27.14    | 26.36   |
>
> As illustrated in Table 1, our TinyLUT-F achieved an average accuracy improvement of over 0.5dB as well as about 5x storage reduction compared to MULUT-SDYEHO-X2. Like the performance validation on color image data in super-resolution task, the quantitative comparisons on color image denoising are still needed for comprehensive assessment. Meanwhile, this additional investigation expands the scope of our work.
>
> **Response to Weakness 3** We fully agree with the reviewer's insightful comment. Following the complexity evaluation in Ref.[7], computational costs of addition and multiplication operations are employed as the metric to assess the model complexity. Following the existing literature, we conduct the complexity comparison on 4x super-resolution task with an output resolution of 1280x720. The results are presented in Table 2. As can be seen, our proposed TinyLUT-S achieves the lowest computational complexity compared to other competitors.
>
> **Table 2 The statistics of addtion and multiplication operations**
> |              | Add    | Mul   |
> |--------------|--------|-------|
> | SRLUT-S      | 44.3M  | 19.4M |
> | MULUT-SDY    | 165M | 69.2M |
> | MULUT-SDY-X2 | 145.5| 73.6M |
> | TinyLUT-S    | 10.28M | 5.3M  |
> | TinyLUT-F    | 46.28M | 26.3M |
>
> To address the reviewer's concern about latency, we conducted 4x super-resolution tests on a Raspberry Pi 4B platform. At present, the mainstream high-resolution images mainly include 1920×1080 (FHD) and 3840×2160 (UHD). Given this, we illustrated the processing time of our TinyLUT and competitive methods in Table 3. As can be seen, TinyLUT-S yields the lowest inference latency, while the latency of TinyLUT-F is comparable to SRLUT-S and far superior to MULUT on high-resolution images. This further validates the high efficiency of TinyLUT on edge devices mentioned by reviewers G5Ti and MFFN.
>
> **Table 3 Latency of high-resolution image on Raspberry Pi 4B**
> |              | 1920x1080(FHD) | 3840x2160(UHD) |
> |--------------|----------------|----------------|
> | SRLUT-S      | 554ms          | 2224ms         |
> | MULUT-SDY    | 853ms          | 3566ms         |
> | MULUT-SDY-X2 | 906ms          | 3834ms         |
> | TinyLUT-S    | 163ms          | 777ms          |
> | TinyLUT-F    | 614ms          | 2797ms         |
>
> **Response to Weakness 4** Thanks for the reviewer's suggestion. Although our article received positive feedback (2kHn) in terms of writing style and chapter division, there are still some grammar issues. We have reviewed the paper again and corrected some errors to improve the quality of the article.
>
> **Response to Question 1**  Thanks for the reviewer's important reminder. We comply with the reviewer's suggestion and replace the Lena image in Fig. 2 with the Baby image in the Set5 test dataset.
> Meanwhile, we confirmed that Lena was not included in the training set. We also removed the Lena image from the Set 14 test dataset and validated the accuracy of each method again. As the PSNR shown in Table 4, the results indicate that after removing Lena image, our TinyLUT still achieves competitive results and does not affect the ranking of our method.
>
> **Table 4 The 4x SR experiment for Set14 without Lena**
> |              | Set14v without Lena | Set14 |
> |--------------|---------------------|-------|
> | SRLUT-S      | 26.80               | 27.01 |
> | SPLUT-L      | 27.26               | 27.54 |
> | MULUT-SDY-X2 | 27.31               | 27.60 |
> | TinyLUT-S    | 27.02               | 27.33 |
> | TinyLUT-F    | 27.72               | 28.01 |
>
> [1] Shaker et al. "SwiftFormer: Efficient Additive Attention for Transformer-based Real-time Mobile Vision Applications", ICCV 2023
>
> [2] Howard et al. "Searching for mobilenet-v3", CVPR 2019
>
> [3] Sandler et al. "Mobilenet-v2: Inverted residuals and linear bottlenecks", CVPR 2018
>
> [4] Woo et al. "ConvNeXt V2: Co-designing and Scaling ConvNets with Masked Autoencoders", CVPR 2023
>
> [5] Liu et al. "A ConvNet for the 2020s", CVPR2022
>
> [6] Zhang et al. "Efficient CNN Architecture Design Guiede by Visualization", ICME 2022
>
> [7] Li et al. "MuLUT: Cooperating Multiple Look-Up Tables for Efficient Image Super-Resolution", ECCV2022

---

> ### Comment · Reviewer_JnS6 · 2024-08-13
> **reply to the author's rebuttal**
>
> Thanks for the authors' reply. Part of my concerns are addressed. Thanks for the efforts of the Ethics reviewers. Although the authors conducted detailed experiments to demonstrate the advantages of the proposed TinyLUT and its variants, I still believe that accelerating the CNN-based image restoration methods is not very convincing, even considering the edge device scenario where domain-specific accelerator (DSA) is still an option. In high-level vision tasks, infering ViT-based models on edge devices has been investigated [1, 2, 3].
>
> [1] Junting Pan, Adrian Bulat, Fuwen Tan, Xiatian Zhu, Lukasz Dudziak, Hongsheng Li, Georgios Tzimiropoulos, and Brais Martinez. Edgevits: Competing light-weight cnns on mobile devices with vision transformers. In European Conference on Computer Vision, 2022
> [2] Abdelrahman Shaker, Muhammad Maaz, Hanoona Rasheed, Salman Khan, Ming-Hsuan Yang, and Fahad Shahbaz Khan. Swiftformer: Efficient additive attention for transformerbased real-time mobile vision applications. In Proceedings of the IEEE/CVF International Conference on Computer Vision, 2023
> [3] Shashank Nag, Logan Liberty, Aishwarya Sivakumar, Neeraja J Yadwadkar, Lizy Kurian John. Lightweight Vision Transformers for Low Energy Edge Inference. ISCA 2024 workshop MLArchSys (Machine Learning for Computer Architecture and Systems).

---

> > ### Author Response · Authors · 2024-08-13
> > **Thanks for the reviewer's reply!**
> >
> > Thanks for the reviewer's reply. We fully agree that ViT in edge device scenarios is a very promising emerging direction! The raised papers indeed demonstrate the growing efforts in breaking into this new field in academia in the last two years.
> >
> >
> > In this NeurIPS submission, we aim to share our recent progress on strategies that we believe, and as also echoed by reviewers, can directly benefit the existing edge computing community where CNN is still the majority.
> >
> >
> > Thanks to the kind suggestion of reviewer JnS6, we are pleased to cite the raised papers in the discussion section, as they not only enrich our manuscript but also serve to highlight and encourage further research efforts on ViT methods for edge devices.

---

> > ### Author Response · Authors · 2024-08-14
> > **Thank you for engaging in the discussion!**
> >
> > We would like to thank the reviewer for engaging in the discussion and for reminding us of the violation regarding the Lena image. We will ensure that all edits mentioned in the rebuttal are incorporated when revising our paper. Thanks again for your participation in the discussion!

---

### Author Rebuttal · Authors · 2024-08-06

We would like to thank all reviewers for their valuable and professional comments. We are thankful that most of the reviewers shared positive feedback for our paper. Meanwhile, we are glad to read that our work was efficient (JnS6, G5Ti, MFFN), effective (G5Ti, MFFN),  valuable (2kHn), potential (G5Ti), well written and organized (2kHn).

We appreciate the reviewers' acknowledgement of our proposed method's potential and contribution to the LUT-based approaches. As reviewer MFFN noted, our study "analyzes the storage explosion challenge of LUT and provides a solution by decomposing the convolution kernel and compressing the quantization scale". Our method solves the storage explosion problem encountered in recently proposed SRLUT[1], SPLUT[2] and MULUT[3]. Our work has promoted the development of LUT-based methods, as expressed by reviewer G5Ti: "The paper demonstrates the great potential of LUT-based methods," and as reviewer 2kHn also expressed: "The proposed method is valuable for advancing image restoration on edge devices."

However, the reviewers have some concerns about our paper. The reviewer expressed concern that our proposed method is only suitable for CNN-based architectures (JnS6). Meanwhile, the reviewers suggest that we should compare with the latest LUT-based methods, especially the SPFLUT[4] mentioned by G5Ti and 2kHn. The last concern is the violation of the Lena image mentioned by reviewer JnsS6.

Regarding the concern that our method is only suitable for CNN-based architectures. Indeed, our proposed TinyLUT as well as existing LUT-based methods are mainly designed for CNN-based architectures. This is because CNN are still the preferred choice for real-time deployment on mobile devices, primarily because the convolution operation is computationally efficient[5,6,7]. CNNs compete favorably with Transformers in terms of accuracy and scalability, and convolution remains much desired and has never faded[8,9]. In recent years, many CNN methods have still been proposed[8,9,10].

Regarding the comparison with the latest LUT-based methods, the [4] proposed SPFLUT+DFC method, was published in CVPR2024 after we submitted the manuscript. To demonstrate the advantages of our method, we have conducted the comparisons on image super-resolution and image denoising tasks with SPFLUT+DFC in Table 1. As can be seen, TinyLUT-F achieves the highest PSNR with about 12x storage reduction compared to SPFLUT+DFC. As described in the 'Strengths' section by G5Ti, these experiments further demonstrate the value of our method for image restoration on edge devices.

**Table 1 Quantitative comparisons on 4x SR task**
|            | storage  | Set5  | Set14 | BSD100 | Manga109 | Urban100 |
|------------|----------|-------|-------|--------|----------|----------|
| SPFLUT     | 17.284MB | 31.11 | 27.92 | 27.10  | 28.68    | 24.87    |
| SPFLUT+DFC | 2.018MB  | 31.05 | 27.88 | 27.08  | 28.58    | 24.81    |
| TinyLUT-F  | 171KB    | 31.18 | 28.01 | 27.13  | 28.83    | 24.92    |

In addition, we compare SPFLUT+DFC and our TinyLUT in image denoising task with noise level 15 on Set12 and BSD68. As shown in Table 2, TinyLUT still yields about 0.2dB higher PSNR with 3× storage reduction compared to SPFLUT+DFC. The results further demonstrate that TinyLUT exhibits notable advantages in other image restoration tasks. These experiments corroborate the effectiveness of the proposed method, as noted by reviewers (G5Ti, MFFN).

**Table 2 Quantitative comparisons on image denoise**
|            | storage | Set12 | BSD68 |
|------------|---------|-------|-------|
| SPFLUT     | 3017KB  | 32.11 | 31.17 |
| SPFLUT+DFC | 595KB   | 32.01 | 31.09 |
| TinyLUT-F  | 187KB   | 32.22 | 31.20 |

Regarding the violation of the Lena image mentioned by reviewer JnS6, we will use the Baby image from the Set5 test set to modify Figure 2 in our paper. We have ensured that Lena was not used in the training process and found that removing it from testing does not affect the conclusion of our paper.

[1] Jo et al. "Practical Single-Image Super-Resolution Using Look-Up Table",  CVPR2021

[2] Ma et al. "Learning Series-Parallel Lookup Tables for Efficient Image Super-Resolution",  ECCV2022

[3] Li et al. "MuLUT: Cooperating Multiple Look-Up Tables for Efficient Image Super-Resolution",  ECCV2022

[4] Li et al. "Look-Up Table Compression for Efficient Image Restoration",  CVPR2024

[5] Shaker et al. "SwiftFormer: Efficient Additive Attention for Transformer-based Real-time Mobile Vision Applications", ICCV 2023

[6] Howard et al. "Searching for mobilenet-v3", CVPR 2019

[7] Sandler et al. "Mobilenet-v2: Inverted residuals and linear bottlenecks", CVPR 2018

[8] Liu et al. "A ConvNet for the 2020s",  CVPR2022

[9] Woo et al. "ConvNeXt V2: Co-designing and Scaling ConvNets with Masked Autoencoders", CVPR 2023

[10] Zhang et al. "Efficient CNN Architecture Design Guiede by Visualization",  ICME 2022

---

### Decision · Program_Chairs · 2024-09-25

**Decision:**

Accept (poster)

**Comment:**

The paper considers lookup table based methods for accelerating CNNs for image reconstruction. The paper notes that the memory complexity of lookup table based methods for image reconstruction grows exponentially in the size of the convolutional kernel, which may create a storage issue.

The paper received four reviews:
* Reviewer JnS6 (3: reject) main concern is that accelerating CNN-based methods (in particular DnCNN) is not convincing and that more modern architectures should be considered. I agree with the authors' response that CNN based methods are still relevant in particular when speed matters. The reviewer also notes that the paper is not well written, and I fully agree to this concern.
* Reviewer G5Ti (6: weak accept after rebuttal) notes that the paper should compare to recent works that address the same problem, in particular the paper
[1] 2024 CVPR paper ``Look-Up Table Compression for Efficient Image Restoration'',
which is about the exact same issue. In the rebuttal the paper compares to the paper [1].
* Reviewer 2kHn is also concerned about a lack of comparisons, that comparisons on more tasks (beyond denoising and super-resolution) are needed, and that the performance in terms of PSNR lacks behind state-of-the-art methods. The authors provided additional comparisons and notes that while the performance is slightly lower than state-of-the-art methods, the latency and storage are much better, which is a valid point. I also think that considering denoising and super-resolution is sufficient.
* Reviewer MFFN (7: accept after rebuttal) has no major concerns after rebuttal.

In summary, the paper proposes a method to effectively reduce the storage complexity of lookup table based methods for accelerating CNNs, and the method is sufficiently validated.